# Metallurgical Characterization of Welded Joint of Nanostructured Bainite: Regeneration Technique versus Post Welding Heat Treatment

**DOI:** 10.3390/ma13214841

**Published:** 2020-10-29

**Authors:** Aleksandra Królicka, Krzysztof Radwański, Aleksandra Janik, Paweł Kustroń, Andrzej Ambroziak

**Affiliations:** 1Department of Metal Forming, Welding and Metrology, Wrocław University of Science and Technology, Wybrzeże Wyspiańskiego 27, 50-370 Wroclaw, Poland; pawel.kustron@pwr.edu.pl (P.K.); andrzej.ambroziak@pwr.edu.pl (A.A.); 2Łukasiewicz Research Network-Institute for Ferrous Metallurgy, K. Miarki 12-14, 44-100 Gliwice, Poland; kradwanski@imz.pl (K.R.); ajanik@imz.pl (A.J.)

**Keywords:** welding, regeneration technique, nanostructured bainite, nanobainite, HAZ, fusion zone, materials characterization, bainite degradation, PWHT, EBSD

## Abstract

One of the main limitations in application of nanostructured carbide-free bainite as a construction material is the difficulty of joining. This research presents a structural characterization of welded joints of medium carbon 55Si7 grade steel after the welding process with a regeneration technique as well as post welding heat treatment (PWHT). The hardness distribution of the welded joint with regeneration exhibit an overall decrease in hardness when compared to the base material and a significant decrease in hardness was observed in the heat-affected zone (HAZ). Unfavorable hardness distribution was caused by the presence of diffusion-type transformations products (pearlite) in the HAZ and bainite degradation processes. On the other hand, welding with the PWHT promotes the achievement of a comparable level of hardness and structure as in the base material. However, a slight decrease in hardness was observed in the weld zone due to the micro-segregation of the chemical composition caused by the indissoluble solidification structure. Based on the structural analysis, it was found that steel with relatively low hardenability (55Si7) should be welded using PWHT rather than a regeneration technique.

## 1. Introduction

Nanostructured (NB) carbide-free (CFB) bainitic steels have been widely developed for many years due to high mechanical properties. The carbide-free structure is possible to obtain by the concentration of approximately 2% wt. Si, which inhibits the precipitation of cementite from austenite [1,2]. The high mechanical properties (tensile strength above 2000 MPa, yield strength at least 1100 MPa, and elongation up to 14% [1,2,3,4,5,6]) are determined mainly by the structural factor consisting of nanometric scale widths of the bainitic ferrite laths (below 100 nm) and film-like austenite (even 20–40 nm [3]). One of the main limitations of industrial applications of NB CFB steels as constructional material is the difficulty of the welding processes [7,8]. Carbon equivalent (CEV or CE) is the most common tool to determine the weldability of steels. CEV is strictly dependent on the chemical composition of the steel and weldability is significantly reduced by an increased content of chemical elements such as C, Mn, Cr, Mo, V, Ni, and Cu. Poor weldability of nanobainitic steels is primarily associated with the chemical composition, which usually contain 0.6–1.0 wt.% C, 1.5–3.0 wt.% Si, and 0.6–2.0 wt.% Mn [9]. In recent years, attempts have been made to join nanostructured carbide-free steels using tungsten inert gas (TIG) [10,11,12,13,14], rotary impacting trailed welding [15], impacting trailed welding [16], laser beam welding [13], and friction stir welding [17] methods. Achieving high mechanical properties requires obtaining a comparable structure in the entire area of the welded joint. For this reason, it was proposed to perform isothermal heat treatment of welded joints after the welding process and to perform the welding process in the as-delivered state (softened) [11]. The heat treatment after welding allowed us to avoid cold cracks in welds. Furthermore, preheating was also used. However, it should be highlighted that this is an additional technological process that cannot always be applied (e.g., due to dimensions of elements) and generates higher costs. Another method to increase the strength of welded joints is the regeneration technique proposed by Fang et al. [12,13,14]. The regeneration technique consists of controlled cooling after the welding process and isothermal annealing at the temperature of the designed bainitic transformation and at a time allowing for the completion of the bainitic transformation. The use of regeneration allows avoiding cold cracks (no martensite in the structure) and obtaining a bainitic structure in the almost entire welded joint. Due to the low temperature of bainitic transformation, which enables us to obtain high mechanical parameters, the regeneration process lasts longer. The reduction of regeneration time is possible by introducing deformation in the welded joint because the bainite transformation time for deformed austenite is shorter [15]. To reduce the time, a Rotary Impacting Trailed Welding method has been proposed [15]. This method involves synchronously welding and impacting processes, which introduce compressive and shear stresses at the same time during the welding process. Another method to reduce regeneration time is a refinement of austenite grains [16] because, in high carbon steels, the transformation time will be shortened [18]. For this purpose, the two-pass Impacting Trailed Welding method was applied [16] where two-stage regeneration with static recrystallization occurs. After welding, the grains are reduced in the impact zone from 106 ± 42 μm to 36 ± 13 μm (in base material, about 50 μm). After welding using this method, a significantly higher tensile strength was achieved in comparison to the TIG method without static recrystallization.

The weakest point of welded joints is the Low-Temperature Heat-Affected Zone (LTHAZ) [8,12,13], where the temperature below the lower transformation temperature (A_1_) is affected. In this zone, the degradation of nanostructured carbide-free bainite occurs. Austenite with blocky morphology is decomposed to fine-dispersive pearlite, while the austenite with film-like morphology is decomposed to ferrite and cementite [19]. However, according to the research [13], the product of austenite decomposition in the Heat-Affected Zone (HAZ) is bainite. In addition, in this zone, a fracture after tensile strength tests and the lowest level of hardness were found [12,13]. The limitation of the degraded bainite with cementite precipitation zone can be achieved, among others, by reducing heat input [13]. In addition, due to the lower heat input, it is possible to reduce the content of adverse blocky austenite formed in the weld [10]. In the context of maintaining high mechanical parameters, besides the selection of the welding method, process parameters are also significant.

The authors described in detail the issue of the welding processes of nanobainitic steels in a review article [8].

In these investigations, an attempt was made to present a detailed evaluation of the structure of the welded joint of medium-carbon 55Si7 grade steel, which is scheduled to obtain nanostructured bainite. To maximize the mechanical properties by obtaining a bainitic structure in the entire joint area, the regeneration technique and post welding heat treatment (PWHT) were used. The evaluation of morphological changes was made based on structure investigations (using scanning electron microscopy (SEM) and electron backscattered diffraction (EBSD) methods) and the hardness measurements. In other research, the focus was primarily on the LTHAZ, while this paper describes in detail all existing zones (LTHAZ, High-Temperature HAZ, Fusion Zone). The structure changes caused by the welding process obtained by these two methods were also compared. Moreover, these investigations provide a better understanding of the bainite degradation processes after welding.

## 2. Materials and Methods

Commercial spring steel 55Si7 in the as-delivered state, and, after isothermal heat treatment designed to obtain nanostructured bainite, is analyzed. The chemical composition of this steel is presented in Table 1. This steel is delivered as a sheet with a thickness of 2 mm and characterized by a medium-carbon (0.57 wt.%) and high-silicon (1.89 wt.%) content. Furthermore, it does not contain a significant content of carbide-forming elements such as V, W, Cr, and Mo, which is important in the context of the limitation of carbide precipitations. The research material in the as-delivered state exhibits a structure typical for softening annealing consisting of cementite in a ferrite matrix. Cementite is mainly characterized by spheroidal morphology with various diameters. Locally, cementite with lamellar morphology (incomplete spheroidization) is also found (Figure 1a). The steel has been heat-treated including isothermal annealing. The heat treatment parameters and hardness after this process are presented in Table 2. The obtained structure consists of nanostructured bainite (also called degenerated upper bainite) [20] (Figure 1b). The width of bainitic ferrite laths are usually about 100–150 nm, whereas austenite with film-like morphology exhibited much smaller dimensions (below 50 nm) (Figure 1c).

The welding process in an as-delivered state was carried out on samples of 2 × 100 × 150 mm, while the samples for regeneration were 2 × 80 × 80 mm (due to technical limitations related to the dimensions of the isothermal heat treatment furnace). The samples were welded by the TIG method with a gas shield (argon) according to the parameters presented in Table 3. For both welding processes, preheating was applied, which is exactly as much as the designed bainitic transformation temperature. Preheating of samples in the as-delivered state (and annealing at this temperature for a short time, later air cooling) was used to avoid cold cracks. However, in the case of the heat-treated state intended for regeneration, preheating was used for higher process stability and temperature control, so as not to exceed the M_s_ temperature. The same process parameters were used in both processes based on which heat input was 0.142 (kJ/mm). Steel in a heat-treated state after welding was immediately transferred to a furnace at a temperature of 300 °C and annealed for 10 h to complete the bainitic transformation. On the other hand, steel in the as-delivered state after welding and cooling to the ambient temperature was subjected to Post Welding Heat Treatment (PWHT). The parameters of PWHT (Table 3) were the same as in the case of the heat-treated state (Table 2) to compare both welded joints. The case of the PWHT heating was also important. The presence of hardening structures in the weld instead of spheroidite may cause susceptibility to hot cracks during heating to a relatively high austenitization temperature. Therefore, a stop at 650 °C/5 min was carried out.

To estimate the maximum temperatures (T_max_) occurring in the selected HAZs of welded joints, a numerical simulation of heat cycles was performed using the Simufact Welding software (V2020, Simufact Engineering GMBH, Hamburg, Germany) (Figure 2). Parameters of the welding process of the numerical simulation were similar to the experimental welding process (Table 3). Preheating was also determined. A modeled sheet used in the numerical simulation was similar to the sheets used in experimental welding processes (2 × 80 × 80 mm and 2 × 100 × 150 mm). The distance between the analyzed points was 0.3 mm, and the first point was placed at the center of the welded joint. At the center point of the weld, the T_max_ was approximately 3200 °C. Based on the heat cycles’ simulation, it was found that the measuring points Point 1, Point 2, and Point 3 corresponded to the fusion zone where solidification occurred. The High-Temperature Heat-Affected Zone (HTHAZ) corresponded to the areas where the T_max_ was higher than the A_3_ temperature (Point 4, Point 5, and Point 6). Thus, complete recrystallization of the structure occurred in the HTHAZ regions. However, point 7 corresponded to partial recrystallization (T_max_ was in the range between temperatures A_1_ and A_3_). This zone was located between the HTHAZ and LTHAZ. Point 8 and Point 9 corresponded to LTHAZ where T_max_ did not exceed temperature A_1_. While Point 10 corresponded to the base material due to the lack of significant influence of the heat affect related to the welding process.

The hardness distributions of the welded joints were made using the Vickers method (according to [21]). Matsuzawa MMT-X7B hardness tester (Matsuzawa, Akita, Japan) was used in this investigation. A load of 9.81 N (which corresponds to 1 kg-HV1) was applied for 15 s. The hardness measurements were performed on the cross-section of the welded joints and the distance between the indentations was approximately 0.5 mm.

Metallographic samples were performed by a standard method including grinding and polishing with diamond paste (ending with an abrasive medium size of 1 μm). The prepared samples were etched by nital (revealing the microstructure) and a 5% solution of picric acid in water at 60 °C (revealing the solidification structure). The microstructure of the welds and base material was investigated using an Eclipse MA200 Light Microscope (LM) ((Tokyo, Japan)) with a Nikon DS-fi CCD camera and using FEI INSPECT F Scanning Electron Microscope (SEM) (FEI Company, Hillsboro, OR, USA). Observations performed by SEM methods were carried out using a secondary electron detector (topographic contrast). The accelerating voltage of 15 kV and a working distance of 10 mm were used in this research. The structure of welds and base material was also analyzed by a scanning electron microscope JEOL JSM-7200F equipped with an electron backscattered diffraction (EBSD) detector (JEOL, Tokyo, Japan). The EBSD investigations were performed in the areas of 45 × 104 μm and the step size of 0.2 μm. Data obtained by the EBSD method was processed using the TSL software (OIM Analysis™ 8, Berwyn, PA, USA). A Confidence Index Standardization (CIS) and single iteration grain dilation “clean-up” procedures were made for each analysis. The clean-up procedure, which had been used, concerned less than 5% of the measurement pixels.

## 3. Results

The description of each weld zone was divided into base material (BM), Low-Temperature Heat-Affected Zone (LTHAZ), High-Temperature Heat-Affected Zone (HTHAZ), and Fusion Zone (FZ), as presented in Figure 3. Selected zones are characterized by different structural morphology and mechanical properties.

### 3.1. Welded Joint with a Regeneration Technique

Samples after welding with the regeneration technique exhibit clear heat-affected zones and weld after etching by nital (Figure 4). The hardness distribution was characterized by a gradual course with a significant decrease in hardness at the LTHAZ. In the area of the entire welded joint, the obtained hardness was significantly lower than the base material before the welding process (547 ± 3 HV1, Table 2). In the FZ zone, hardness in the range of 451–473 HV1 was noted. In the HTHAZ, the hardness gradually decreased when compared to FZ. On the boundary of the LTHAZ and HTHAZ, the largest decrease in hardness (about 320 HV1) was obtained, and then, toward the base material, slowly increased. It must be pointed out that the hardness of the base material after welding with the regeneration technique (about 410 HV1) was lower than before the welding process. This proves that the regeneration technique also affected the base material, which requires validation by structural investigations. To evaluate the structure, areas “A,” “B,” “C,” “D,” and “E” were specified (Figure 4), which were subjected to structural morphology analysis in the following sections.

#### 3.1.1. Fusion Zone Characterization

In the FZ, the presence of a dendritic solidification structure typical for welding processes was found (Figure 5a). After etching by nital, a structure was revealed (Figure 5b and Figure 6). In the dendrite areas, the lath morphology of degenerated upper bainite consisting of coarse bainitic ferrite and austenite was found. The structure is more coarse when compared to the base material. Austenite with blocky morphology has also been identified especially in the interdendritic areas (Figure 6c), similar to References [14,22]. The presence of martensite was excluded. The obtained structure in this zone meets the requirements of the regeneration technique because the bainitic structure was obtained. On the other hand, heterogeneity caused by dendritic structure determines important morphological changes. Despite the required structure, its morphology reduces hardness and, thus, mechanical properties.

Figure 7a shows locally, in some of the tested areas, different orientations of bainitic ferrite were present. In the dendrite areas, austenite exhibited only one orientation (Figure 7a) because it has the same orientation as the parent austenite. The morphology of austenite is partly lath. Locally, austenite occurred with blocky morphology and formed islands, especially on dendrite boundaries (Figure 7b). Crystallographic orientations between ferrite and austenite are described by four main relationships (OR): Bain OR [23], Nishiyama-Wassermann (N-W) OR [24], Kurdjumov-Sachs (K-S) OR [25], and Greninger-Troiano (G-T) OR [26]. It follows that each ferrite variant can form a set of 24 different parent austenite orientations for the K-S and G-T ORs, 12 for the N-W OR, and three for Bain OR. The crystallographic relations between ferrite and austenite were consistent with the K-S and N-W orientations (Figure 7c). However, the K-S orientation dominated more than N-W. The dominance of K-S orientation was also found in Reference [27] for higher temperature ranges of bainitic transformation. Distribution of misorientation angles exhibited the presence of two modes in the area of high-angle boundaries (HABs). Angles in the 42–47° range prevailed, which refer to the ideal N-W (ideal angle: 45.99°) and K-S (ideal angle: 42.85°) orientations. An angular accuracy of approximately ± 1° was used in performed EBSD analysis (Table 4). The orientation close to ideal N-W OR usually dominates for carbide-free nanostructured bainite [28]. For lower bainite and tempered martensite, the pair of misorientation angles of adjacent packets typically exhibits 55° to 60° [29,30]. In contrast, also 12 variants and 5 misorientation angles are possible for N-W orientation (13.76°, 19.47°, 50.05°, 53.69°, and 60°) [31]. The existing second mode (approximately 55°) may also point to the boundary between ferrite laths and austenite. For this reason, unambiguous structure identification also requires the investigations of the structure morphology of tested welded joint. A low fraction of low-angles boundaries (LABs) was identified (Figure 7d). Therefore, based on the distribution of misorientation angles, it became evident that distribution in the FZ was consistent with the distributions typical for bainitic steels.

#### 3.1.2. High-Temperature Heat-Affected Zone

In the HTHAZ according to Figure 4, two areas were distinguished: coarse-grained “B” and fine-grained “C” in which the structure is presented in Figure 8. These zones were characterized by significantly different hardness levels. The hardness value in the area “B” was about 425 HV1, while, in the “C” area, about 375 HV1. The structure in the “B” area mainly consisted of degenerated upper bainite (Figure 8a). It was also found that a low fraction of austenite with block morphology is present in this area (Figure 9a,b). Bainitic ferrite laths also had a varied width, similar to austenite. Based on the size of the structure constituents, it was estimated that the grain size of the prior austenite grains was varied (unstable grain growth). A different structure was identified in the “C” area, which, besides degenerated upper bainite, contained diffusion-type transformation products (Figure 8b). The structure in this area consisted of degenerated upper bainite, blocky austenite, and pearlite (Figure 9c). The identified HTHAZ perlite was characterized by a fine-dispersive structure (Figure 9d). Moreover, the fraction of pearlite increased with the distance from the end of FZ, which is also shown by a decrease in hardness.

The EBSD investigations were conducted for both analyzed zones: “B” (Figure 10) and “C” (Figure 11). It appears that the structure of these zones differs significantly, which was also confirmed by SEM observations. In the “B” area (Figure 10), the structure consisted of degenerated upper bainite and blocky austenite. The orientations of bainitic ferrite laths varied in the area of prior austenite grains, whereas austenite usually exhibited the dominant orientation inside the grain (Figure 10a). The Phase Distribution map confirmed the presence of bainitic ferrite and austenite with typical morphology for degenerated upper bainite (Figure 10b). Fine blocky austenite was found. K-S orientation was dominant (Figure 10c). The distribution of misorientation angles was characterized by the presence of one dominant mode in the range of the ideal K-S and N-W orientations (Figure 10d). A low fraction of the LABs were identified. In general, the misorientation angles distribution of the “B” (HHAZ) was comparable to the distribution of the FZ (Figure 7d).

However, in the case of zone “C,” the presence of degenerated bainite, pearlite, and blocky austenite was found. In contrast to the area “B” (HTHAZ) and the FZ, bainitic ferrite laths exhibited the predominant orientation of the area of prior austenite grains (Figure 11a). The pearlitic ferrite orientations were visible and also exhibited varied orientations between pearlite colonies. As in Reference [32], it was found that more than one colony can occur inside the one pearlite nodule. It was also found that pearlite is fine-dispersive and located on the boundaries of the prior austenite grains. However, an unambiguous colony size assessment cannot be determined. In the area of degenerated upper bainite, austenite with blocky morphology existed (Figure 11b). Blocky austenite usually occurred in the pearlite colonies. The distribution of misorientation angles also exhibited a mode in the range corresponding to K-S and N-W ORs (Figure 11d). It must be pointed out that, for the fine pearlitic structures, the magnification should be higher (and, thus, the resolution) to correctly determine the occurring misorientation angles, which is associated with the presence of lamellar cementite. Therefore, in the case of a multi-phase structure, SEM observations and the EBSD method should be complementarily analyzed.

#### 3.1.3. Low-Temperature Heat-Affected Zone and Base Material

The highest decrease in hardness was observed at the boundary of the HTHAZ and the LTHAZ (the area between “C” and “D,” marked in Figure 4). In this area, degraded bainite (caused by exposure to a temperature lower than $A_{1}\;$during the welding process) and pearlite was identified (Figure 12a and Figure 13a). The degraded bainite consisted of bainite ferrite laths, cementite precipitates mainly inside the laths, and partially decomposed austenite (Figure 13a). The pearlite present in this zone exhibited larger inter-lamellar spaces and was less dispersive in comparison to the “C” area (HTHAZ). The fraction of pearlite decreased toward the base material and was not identified in area “D” (LTHAZ). In the “D” area, where hardness gradually increased, only degraded bainite was found (Figure 12b and Figure 13b). However, the degree of bainite degradation was higher than in the previous area. This structure consisted of bainitic ferrite and cementite precipitates both inside and at the boundaries of laths. The presence of cementite at the boundaries of prior austenite grains was also found (Figure 13b). Then, the degree of bainite degradation decreased toward the base material. The structure of base material (area “E”) consisted of a degenerated upper bainite, lower bainite, and a low fraction of blocky austenite (Figure 12c and Figure 13c). The lath morphology of the structure is more clearly visible when compared to the previous zone (area “D”). It should be highlighted that the hardness obtained in this zone is lower than one of the material before the welding process. In addition, in this zone, the cementite precipitations occurred, which indicates the effect of enough high temperature for early degradation of the structure. Besides, prolonged reheating at the transformation temperature during the regeneration technique may also be the reason for the precipitation processes.

The EBSD investigations were performed for both analyzed zones: “D”-LTHAZ (Figure 14) and “E”-base material (Figure 15). It appears that the structure of these zones differs significantly, which was also confirmed by prior SEM observations. In the “D” area (Figure 14), the structure consisted of a degraded bainite structure. Austenite in this area was completely decomposed (Figure 14b). This means that the austenite in this zone was decomposed into ferrite and cementite. The orientations of bainitic ferrite laths varied in the area of prior austenite grains (Figure 14a). Due to the complete decomposition of austenite, no significant fraction of the K-S and N-W orientations was found (Figure 14c). The distribution of misorientation angles was characterized by the dominant presence of angles in the range of 50–60°, which may indicate a significant fraction of lower bainite. A high fraction of the LABs were identified, which proves the processes of cementite precipitation in ferrite subgrains. A similar distribution of misorientation angles was also found in the LTHAZ of the bainite rail after the welding process [33].

The “E” area (BM), was subjected to the structure analysis using the EBSD method (Figure 15). The lath morphology of the bainitic ferrite was mainly preserved (Figure 15a). However, it is visible that this morphology is also partially decomposed. Based on the phase distribution map, it was found that austenite content was reduced (Figure 15b). For this reason, only locally the K-S and N-W misorientation angles were found (Figure 15c). The distribution of misorientation angles (Figure 15d) exhibited significant differences when compared to previously analyzed zones of welded joints with regeneration techniques. The mode occurs here for the range of 47–60°, which corresponds to the boundaries between bainitic ferrite laths [29]. Besides, a high fraction of low-angle boundaries were found in the range of 5–15°. A misorientation angle in the range of 2.5–8° can be explained by the presence of lath-like sub-grains of ferrite [34], which were likely caused by the decomposition of austenite. Thus, the structure of the base material for the regeneration technique also exhibits some symptoms of the early stage of bainite degradation, but less severely than in the case of LTHAZ (Figure 14).

#### 3.1.4. Comparision of Blocky Austenite Content

Based on the EBSD analysis, changes in the austenite fraction in the zones of welded joints were determined (Table 5). Mainly blocky austenite was identified, while austenite with film-like morphology was not identified due to its nanometric dimensions. Blocky austenite determines the plastic properties of the material. Thus, its fraction is significant in the context of designing welding processes. The highest fraction of austenite (about 26%) was found in the case of the HTHAZ C region, where partial austenitization has occurred. An increased austenite fraction was also found in HTHAZ region B (about 19.4%) and the fusion zone (about 9.3%). Complete decomposition of austenite was found in LTHAZ. The fraction of austenite (blocky) in the base material was about 1.4%. In general, the fraction of the N-W and K-S ORs misorientation angles is related to the amount of identified austenite. It can be concluded that, in the case of all zones of welded joints (excluding base material), the K-S orientation rather than the N-W orientation dominated.

### 3.2. Welded Joint with Post Welding Heat Treatment

Samples after welding in an as-delivered state and PWHT processes do not exhibit a visible weld area after etching by nital and picric acid solution (Figure 16). The hardness distribution was characterized by a gradual course with comparable values for the weld and base material. The hardness of the base material was the same as the one of the heat-treated material without welding processes. The hardness of the base material was comparable to heat-treated 55Si7 steel without the welding processes (Table 2). A slight decrease in hardness (about 15–25 HV1) was noted for the fusion zone. The decrease in hardness may result from slight differences in the structural morphology. Due to the character of the hardness distribution and the preliminary structure assessment, three areas were selected for further investigations: “A” corresponding to FZ, “B” corresponding to HAZ, and “C” corresponding to BM (Figure 16).

#### 3.2.1. The Fusion Zone and Heat-Affected Zone

In area “A” (FZ) and area “B” (HAZ), the structure consisted mainly of bainite with lath morphology (Figure 17). For the FZ, HAZ, and BM, no significant differences in prior austenite grain size were found. In the FZ, the presence of a dendritic solidification structure typical for welding processes was found after etching by saturated picric acid solution (Figure 17a). However, it is worth noting that the dendrites are as pronounced as in the case of the welding with the regeneration technique (Figure 5b). This proves that, despite the PWHT process, the solidification structure from the welding process was not completely dissolved. However, the structure revealed in this area (Figure 17b) did not exhibit similar morphology as in the case of FZ after welding with the regeneration technique (Figure 5a). Additionally, the structure in the “A” area (FZ) was comparable to the structure of the “B” area (HAZ). Mainly degenerated upper bainite and lower bainite occurred in the tested zones (Figure 18). In area “A,” slightly thicker widths of bainitic ferrite laths were found when compared to area “B.”

The “A” area (FZ) was subjected to the structural analysis using the EBSD method (Figure 19). The morphology of bainitic ferrite is mainly lath and exhibits different orientations inside grains (Figure 19a). However, it should be noted that, in some grains, the lath structure of ferrite is disturbed. This can be explained by the segregation of the chemical composition that was created by the insoluble solidification structure (Figure 17b). In these areas, a higher fraction of blocky austenite was observed (Figure 19b). In general, austenite exhibited a blocky morphology. However, the presence of austenite with film-like morphology cannot be excluded because its dimensions are beyond the spatial resolution limit of the SEM/EBSD method. This may also be indicated by the distribution of misorientation angles (Figure 19d). The local mode of distribution is about 55–60°, which may indicate the advantage of lower bainite over degenerated upper bainite. On the other hand, based on the SEM observations, it appears that the lower bainite is present in lower content than the degenerated upper bainite (Figure 18a,b). The boundaries with the misorientation angles typical of this bainite morphology (K-S and N-W ORs) occurred only in the areas of blocky austenite occurrence (Figure 19c), which is understandable. The relatively high fraction of low-angle boundaries may also indicate the presence of an additional phase (austenite with film-like morphology) that cannot be identified by this method. In the case of the presence of refined phases, the quantitative analysis of misorientation angles can be unreliable [27].

#### 3.2.2. The Base Material

The degenerated upper bainite and lower bainite were found in the base material after the welding process and the PWHT (Figure 20). A similar morphology of bainite was also confirmed for 55Si7 steel after isothermal heat treatment at work [35]. The hardness level in the “C” area (BM) also corresponded to the hardness of the material after heat treatment without the welding process (Table 2). Therefore, no changes in the structure after the PWHT process were found when compared to classical heat treatment.

The “C” area, which corresponds to the base material, was subjected to the structural analysis using the EBSD method (Figure 21). The morphology of bainitic ferrite is lath and exhibits different orientations inside prior austenite grains (Figure 21a). The austenite with blocky morphology was identified (Figure 21b). The fraction of areas with the blocky austenite was slightly lower than in the FZ after welding and PWHT (Figure 19b). In contrast, blocky austenite with larger dimensions appeared in the base material, which may indicate that the bainitic transformation was not fully completed. Based on the analysis of 55Si7 steel after similar heat treatment parameters, the occurrence of austenite with film-like morphology was found (TEM method), which was not identified by the EBSD method [35]. This can also be demonstrated by the relatively high fraction of LABs (Figure 21d). The boundaries with the misorientation angles characteristic of this bainite morphology (K-S and N-W ORs) occurred only in the areas of blocky austenite (Figure 21c), which is similar to the FZ. It was also found that the distribution of misorientation angles is similar to the FZ and exhibits a local mode in the range of 55–60°. The smaller number of K-S and N-W misorientation angles results from the fact that only blocky austenite was identified. Based on the complementary SEM and EBSD investigations, it was found that the structure consisted mainly of degenerated upper bainite and a smaller fraction of lower bainite. Thus, the structure morphology of the base material is similar to the FZ.

#### 3.2.3. Comparison of Blocky Austenite Content

Based on the EBSD analysis, changes in the austenite fraction in the zones of welded joints with the PWHT were determined (Table 6). It should be highlighted that mainly blocky austenite was identified, while austenite with film-like morphology was not identified due to its nanometric dimensions. The fraction of austenite in the fusion zone (3.8%) and the base material (3.1%) was comparable. The fraction of the N-W and K-S ORs misorientation angles were generally low and the K-S orientation dominated.

## 4. Discussion

To evaluate the welding technology of nanobainitic steel using the regeneration technique and the PWHT, an analysis of structural changes was carried out, supported by hardness distributions. Significant morphological changes were found for the regeneration technique.

In the regeneration technique, the FZ was characterized by a typical factor for welding a dendritic solidified structure. High-silicon bainitic steels exhibit micro-segregation of the chemical composition of alloying elements during solidification. Micro-segregation significantly affects the kinetics of bainitic transformation, where, for high-alloyed (interdendritic) areas, the transformation will occur more slowly than in the case of low-alloyed areas [36]. In this conducted research, it became evident that blocky austenite was mainly distributed in the interdendritic areas, which was also indicated in References [14,22,36]. Additionally, an increased share of blocky austenite (9.3%) was found when compared to the base material (1.4%). In the dendrite area, degenerated upper bainite with the dominant bainitic ferrite laths orientations was found. The structure obtained in this zone was coarse, which also affected the decrease in hardness (451–473 HV1) compared to the base material before welding (547 ± 2.5 HV1). Thus, despite the obtained bainitic structure, the influence of segregation and size of the microstructural constituents has a negative effect on the mechanical properties of the welded joint. In coarse-grained HTHAZ (area “B”), the presence of mainly degenerated upper bainite and high content of austenite (19.4%) were found. The hardness in this area was slightly lower than in the case of FZ. The austenite with film-like and blocky morphology occurred in this region. In this zone, austenitization during welding was carried out at an extremely high temperature, which affects the content and morphology of austenite. It was found that, for high austenitization temperatures, the content of blocky morphology is increased [37], which is also confirmed by the obtained results. Additionally, in this zone, the bainitic ferrite laths and austenite were more coarse than in the case of the base material, which also affected the obtained hardness. The obtained structures in the FZ and the HTHAZ confirm the completion of bainitic transformation and, thus, the regeneration time had enough time.

The decrease in the hardness of the welded joint with regeneration in the HTHAZ (area “C”) is associated with the appearance of pearlite. The content and dispersion of pearlite increases to the boundary of the LTHAZ and the HTHAZ. The increase in the interlamellar spacing of pearlite is associated with the cooling time, which increased with the distance from the weld face. The presence of pearlite indicates the insufficient hardenability of tested steel, which is a critical factor in the success of welding with the regeneration technique. The lowest level of hardness was noted on the boundary of LTHAZ and HTHAZ where both pearlite and degraded bainite occurred. This proves that this zone was exposed to the temperature between A_1_ and A_3_ during the welding process (incomplete austenitization). The short time of incomplete austenitization did not provide homogeneous austenite. The bainite remaining in this zone is the result of under-heating during this process, which is then degraded during the regeneration technique. The obtained structures in the HTHAZ did not meet the regeneration technique assumptions in which the purpose is to obtain a bainitic structure in the area of the entire welded joint [12,13,14]. Bainite degradation in the LTHAZ is analogous to the stages of tempering processes of nanostructured bainitic steels. The time of the tempering process and exposure to temperature during the welding process is significantly different, which does not allow a direct comparison. However, the presence of similar mechanisms of bainite degradation to tempering processes [19,38,39,40] was found, such as the cementite precipitation and the austenite decomposition. It must be pointed out that the base material for the regeneration technique was characterized by a lower hardness (more than 100 HV1) than the base material before the welding process. The symptoms of the early stage of bainite degradation were also found in this zone. This can be explained by the fact that the area “E” (marked as the base material) was exposed during welding to a temperature higher than the regeneration technique, which also caused morphological changes in the structure. Then the “E” area could also be classified as a low-temperature heat affected zone (with a low degree of bainite degradation). Another hypothesis may be the start of precipitation processes during long-term annealing at 300 °C. It is known that silicon delays precipitation processes at elevated temperatures [40]. However, the regeneration process was carried out at a relatively low temperature but for a long time (longer than required to complete the transformation). It can be presumed that long annealing at the transformation temperature (after the bainitic transformation is completed) may result in less stability of the structure and slow diffusion-type precipitation and decomposition processes. In addition, a high fraction of austenite with film-like morphology may cause earlier precipitation of cementite because carbon-saturated austenite with film-like morphology will be less stable in terms of precipitation processes [41]. In addition, prolonged heating can cause relaxation of stresses in the material formed during the bainitic transformation, which will reduce the impact of the reinforcement caused by the low-temperature transformation. Moreover, for similar heat treatment parameters of 55Si7 steel, the amount of austenite (both blocky and film-like) obtained by XRD analysis was about 6.4% wt. [35]. The result obtained in this research (1.4%) confirms that most of the austenite in the base material exhibited film-like morphology and/or it was partially decomposed during prolonged regeneration at a low temperature.

The second process after welding with the PWHT was carried out, including the conventional isothermal heat treatment of welded joints, to obtain nanostructured bainite. Heat treatment after welding processes is often used to improve the mechanical properties of high-strength steels. A similar process was carried out in the work [11], which proposed heat treatment after the welding process of high-carbon bainitic steel. After complete austenitization of the welded joint, the structure throughout the entire area should consist of nanostructured bainite with similar morphology. A gradual distribution of hardness and its slight decrease in the weld area seems to be an acceptable result. It was confirmed that the morphology of bainite in the area of the welded joint is similar, and the structure consists of degenerated upper bainite, lower bainite, and a low fraction of blocky austenite (3.1–3.8%). The presence of lower bainite may indicate that the silicon content is too low in relation to the carbon content or maybe it could be the result of heterogeneous austenite during heat treatment. Additionally, in the work on 55Si7 steel [35], besides degenerated upper bainite, lower bainite was also found. For the used heat treatment parameters, the dendritic solidified structure was not completely dissolved, which was revealed after etching by a picric acid solution. It follows that the austenitization time of welded joints should be much longer than in the standard approach. The hardness after welding with the PWHT was comparable to the state after isothermal heat treatment without welding processes.

Considering microstructural factors and hardness distributions (Figure 22), the welding process with PWHT allows us to obtain the expected quality of welded joints of nanostructured bainite. However, it should be highlighted that additional heat treatment after welding can increase the cost of joining technology and cannot always be carried out. On the other hand, this method allows achieving comparable mechanical properties and structure to the base material, which is impossible with other methods proposed in the literature. The regeneration technique for the tested 55Si7 steel did not allow obtaining the desired structure and mechanical properties, which results in poor hardenability. The 55Si7 steel contains a relatively low concentration of chromium (0.15% wt.) and manganese (0.69% wt.) responsible for the level of the hardenability. The tendency to reduce Cr and Mn in nanobainitic steels is visible for the second generation of NANOBAIN steels [9]. The lower content of these alloy elements is introduced instead of Al and Co to accelerate bainitic transformation [42]. The level of hardenability is designed for the cooling medium during heat treatment. Thus, the chemical composition of 55Si7 steel is appropriate to obtain nanostructured bainite in this approach (liquid tin bath). However, in the terms of welding with the regeneration technique, the hardenability is too poor in the analyzed experimental conditions (as shown by the high fraction of pearlite in the HTHAZ). It should be highlighted that welding with regeneration requires the use of alloys with high hardenability because, under welding conditions, the cooling rate is lower than in the case of conventional heat treatment cooling medium.

## 5. Conclusions

The following conclusions were formulated from the structural characterization of welded joints with the regeneration technique and post welded heat treatment process.

The critical requirement for obtaining the welded joints of nanostructured bainitic steels with high mechanical properties (comparable to base materials) is the hardenability criterion. For low-manganese and low-chromium (among others 55Si7) steels, after the regeneration technique, the diffusion-type transformation products (fine-dispersive pearlite) occurred in the heat-affected zones. The presence of pearlite significantly reduces hardness in the heat-affected zones. Besides, the regeneration technique generally reduces the hardness of the welded joints compared to the base material without the welding processes. Hardness reduction is caused by the cementite precipitation processes and a decrease in bainitic ferrite reinforcement (relaxation of compressive stresses caused by low-temperature transformation before the welding process).After the welding process with the PWHT, the welded joint exhibited a bainitic structure (degenerated upper bainite, lower bainite, and a low fraction of blocky austenite). A slight reduction in hardness was noted in the weld zone (FZ), which is caused by the micro-segregation of the chemical composition in this zone.To maintain the high mechanical properties of 55Si7 welded joints, the PWHT should be used rather than a regeneration technique. To maximize the mechanical properties, austenitization of welded joints (with the PWHT) should be sufficiently long to dissolve the solidification structure. Preheating should also be used to avoid cold cracks.Generally, the K-S orientation of austenite (mainly block) and bainitic ferrite dominated for the applied heat treatment parameters (PWHT) and regeneration techniques.In the HTHAZs and the fusion zone after the welding process with the regeneration technique, a significant increase in the fraction of blocky austenite was found. On the other hand, the austenite decomposed completely (into cementite and ferrite) in the LTHAZ.The usage of EBSD methods and the SEM observations allow the identification of the bainite morphology in the welded joints. It needs to be highlighted that the SEM and EBSD methods should be used complementarily to ensure correct interpretation. On the other hand, the identification of highly refined phases (e.g., film-like austenite) should be aided by the TEM method.

## Figures and Tables

**Figure 1 materials-13-04841-f001:**
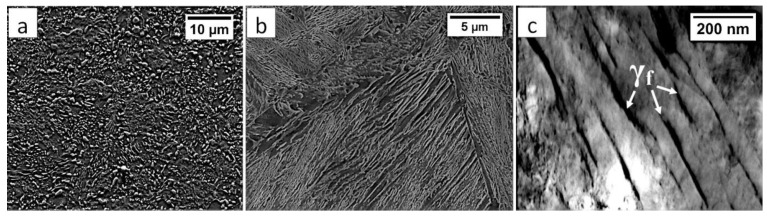
The structure of 55Si7 steel (**a**) in the as-delivered state, and (**b**,**c**) after heat treatment for the regeneration technique. (**a**) Spheroidal cementite with ferrite matrix. Locally visible cementite with lamellar morphology, SEM. (**b**) Degenerated upper bainite with lath morphology, SEM. (**c**) Nanostructured bainite. Laths of ferritic bainite and film-like austenite (γf). TEM.

**Figure 2 materials-13-04841-f002:**
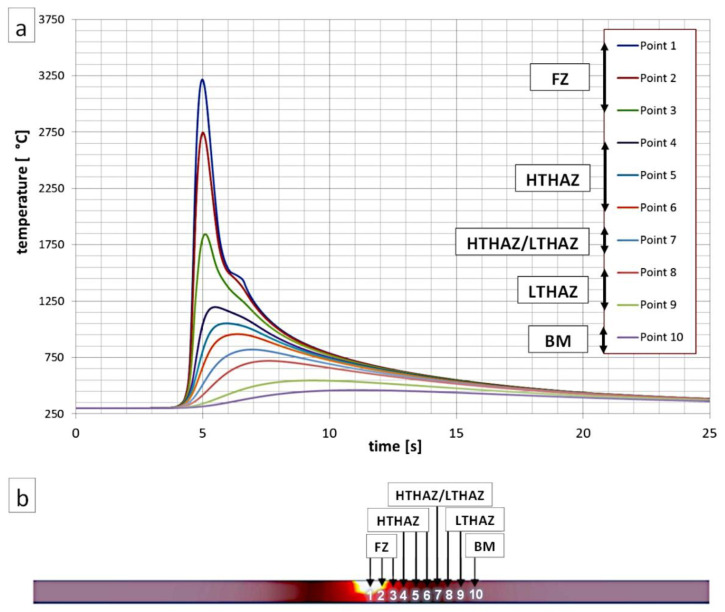
(**a**) Numerical simulation of thermal welding cycles for selected points (indicated in Figure 2b). (**b**) Visualization of the cross-section of the welded joint during the occurrence of maximum temperatures.

**Figure 3 materials-13-04841-f003:**
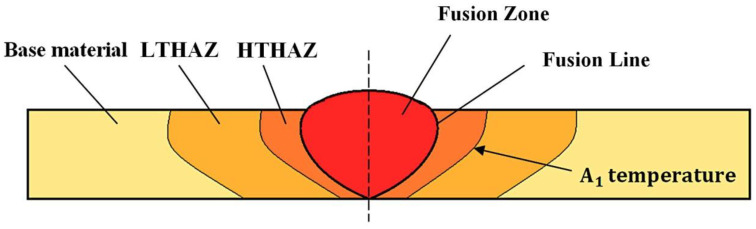
Scheme of the welded joint with specified zones [8].

**Figure 4 materials-13-04841-f004:**
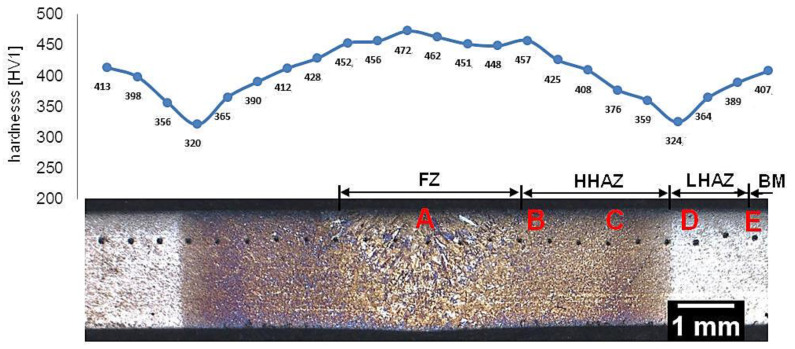
Macroscopic view of the welded joint of 55Si7 steel with regeneration along with the hardness distribution and areas of structural investigations. Etched by nital, LM.

**Figure 5 materials-13-04841-f005:**
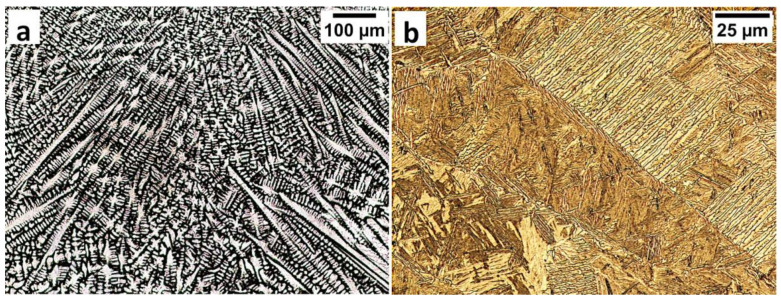
The structure of the fusion zone in the area “A” (marked in Figure 4). (**a**) The columnar, dendritic solidify structure. Etched by saturated picric acid solution, lm. (**b**) Lath morphology of bainite. Etched by nital, LM.

**Figure 6 materials-13-04841-f006:**
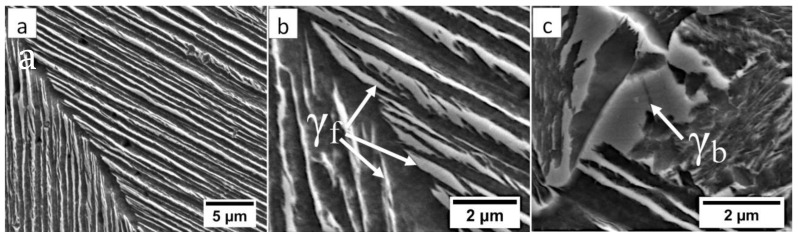
The structure of the fusion zone in the area “A” (marked in Figure 4). (**a**) Lath morphology of coarse bainite inside dendrites. (**b**) Magnification of the area is presented in Figure 6a. Lath morphology of bainitic ferrite and filmy austenite (γ_f_). (**c**) Area of blocky retained austenite (γ_b_). Etched by nital, SEM.

**Figure 7 materials-13-04841-f007:**
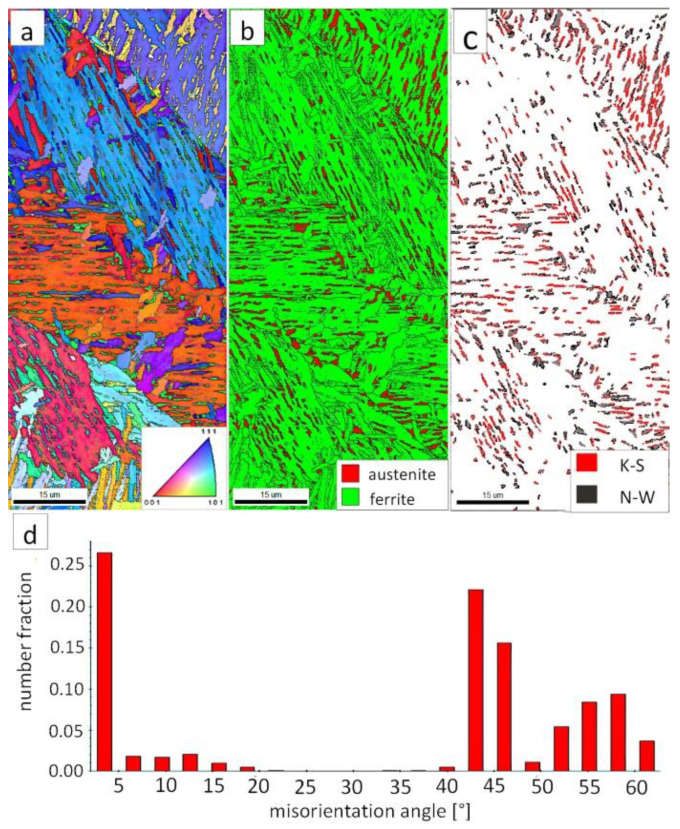
(**a**) Inverse Pole Figure (IPF) Map of the FZ. (**b**) Phase Distribution Map of the FZ. (**c**) Phase distribution map with indicated boundaries of N-W (marked in black) and K-S (marked in red) orientations. Ferrite-white, austenite-gray. (**d**) Misorientation angles distribution of the FZ.

**Figure 8 materials-13-04841-f008:**
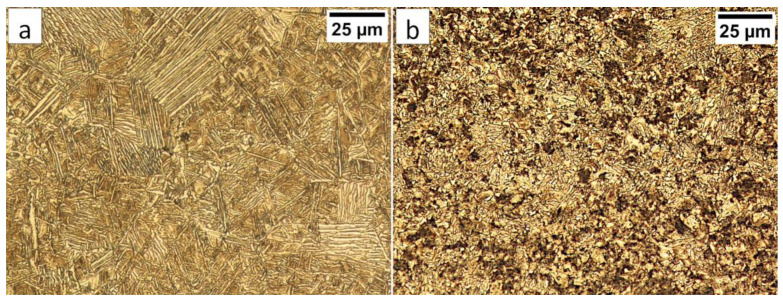
(**a**) The structure of the HTHAZ in the area “B” (marked in Figure 4). Lath morphology of bainite. (**b**) The structure of the HTHAZ in the area “C” (marked in Figure 4). Banite and pearlite. Etched by nital, LM.

**Figure 9 materials-13-04841-f009:**
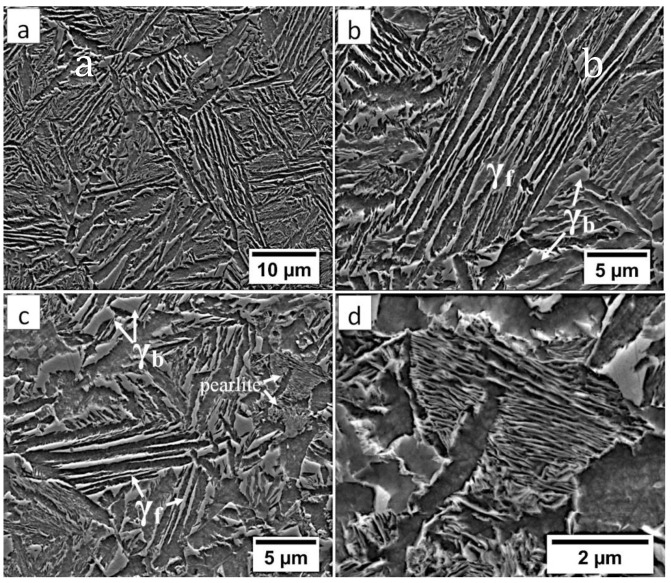
(**a**) The structure of the HTHAZ in the area “B” (marked in Figure 4). Lath morphology of bainite and blocky austenite. (**b**) The structure of the HTHAZ in the area “B” (marked in Figure 4). Different dimensions of bainitic ferrite laths and austenite with filmy (γ_f_) and blocky (γ_b_) morphology. (**c**) The structure of the HTHAZ in the area “C” (marked in Figure 4). Degenerated upper bainite (filmy austenite-γ_f_), blocky austenite (γ_b_), and pearlite. (**d**) The structure of the HTHAZ in the area “C” (marked in Figure 4). Pearlite with fine interlamellar spacing. Etched by nital, SEM.

**Figure 10 materials-13-04841-f010:**
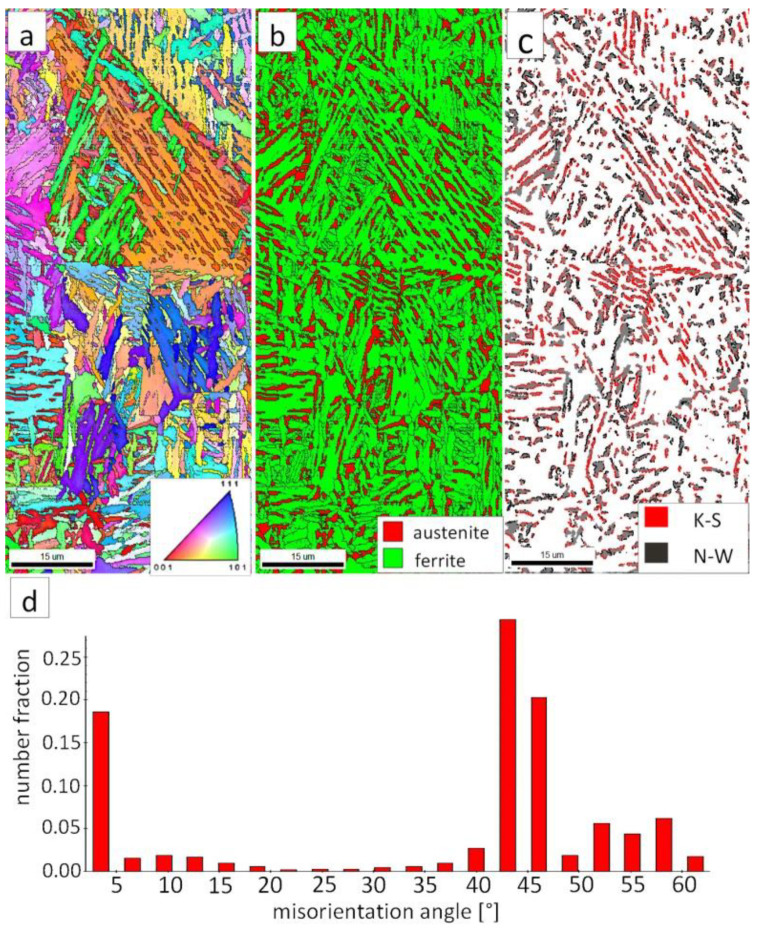
(**a**) IPF Map of the HTHAZ area “B”. (**b**) Phase Distribution Map of the HTHAZ area “B”. (**c**) Phase distribution map with indicated boundaries of N-W (marked in black) and K-S (marked in red) orientations. Ferrite-white, austenite-gray. (**d**) Misorientation angles distribution of the HTHAZ region “B”.

**Figure 11 materials-13-04841-f011:**
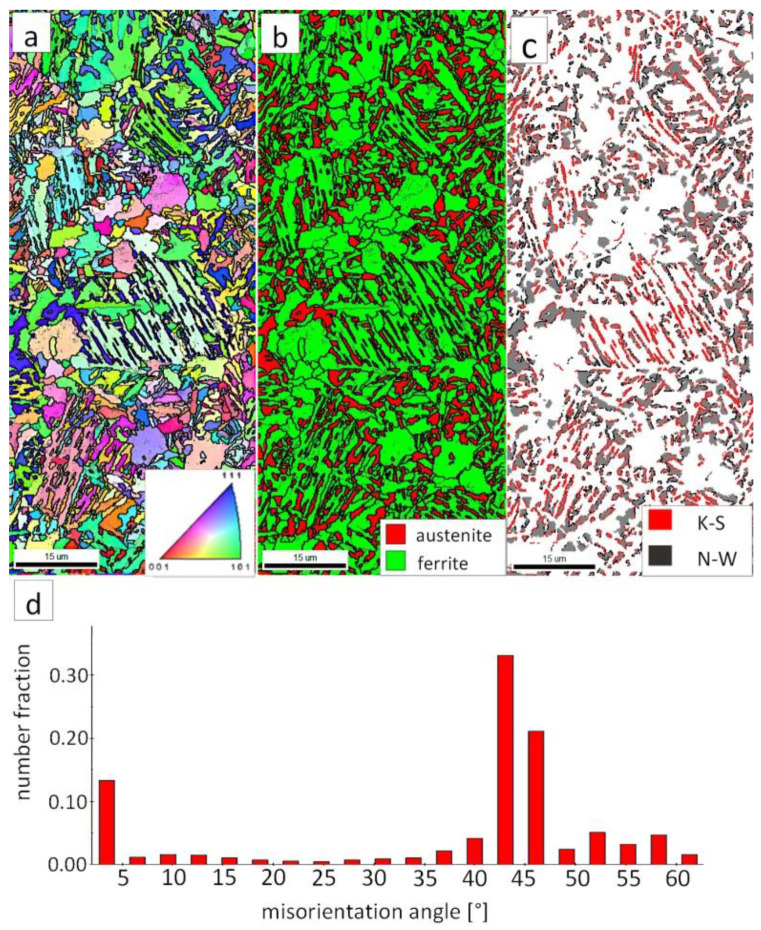
(**a**) IPF map of the HTHAZ area “C”. (**b**) Phase distribution map of the HTHAZ area “C”. (**c**) Phase distribution map with indicated boundaries of N-W (marked in black) and K-S (marked in red) orientations. Ferrite-white, austenite-gray. (**d**) Misorientation angles distribution of the HTHAZ region “C”.

**Figure 12 materials-13-04841-f012:**
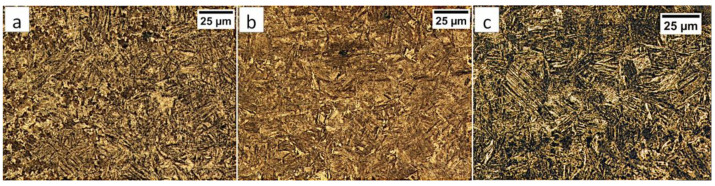
(**a**) The structure of the boundary between HTHAZ and LTHAZ in the area “C/D” (marked in Figure 4). Bainite with cementite precipitations and pearlite. (**b**) The structure of the LTHAZ in the area “D” (marked in Figure 4). Degraded bainite. (**c**) The structure of the base material in the area “E” (marked in Figure 4). Degenerated upper bainite. Etched by nital, LM.

**Figure 13 materials-13-04841-f013:**
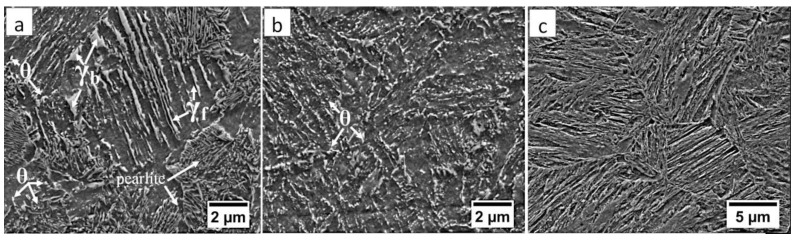
(**a**) The structure of the boundary between HTHAZ and LTHAZ in the area “C/D” (marked in Figure 4). Bainite with cementite precipitations (θ), blocky (γ_b_), and filmy (γ_f_) austenite, and pearlite. (**b**) The structure of the LTHAZ in the area “D” (marked in Figure 4). Degraded bainite with cementite precipitations (θ). (**c**) The structure of the base material in area “E” (marked in Figure 4). Degenerated upper bainite. Etched by nital SEM.

**Figure 14 materials-13-04841-f014:**
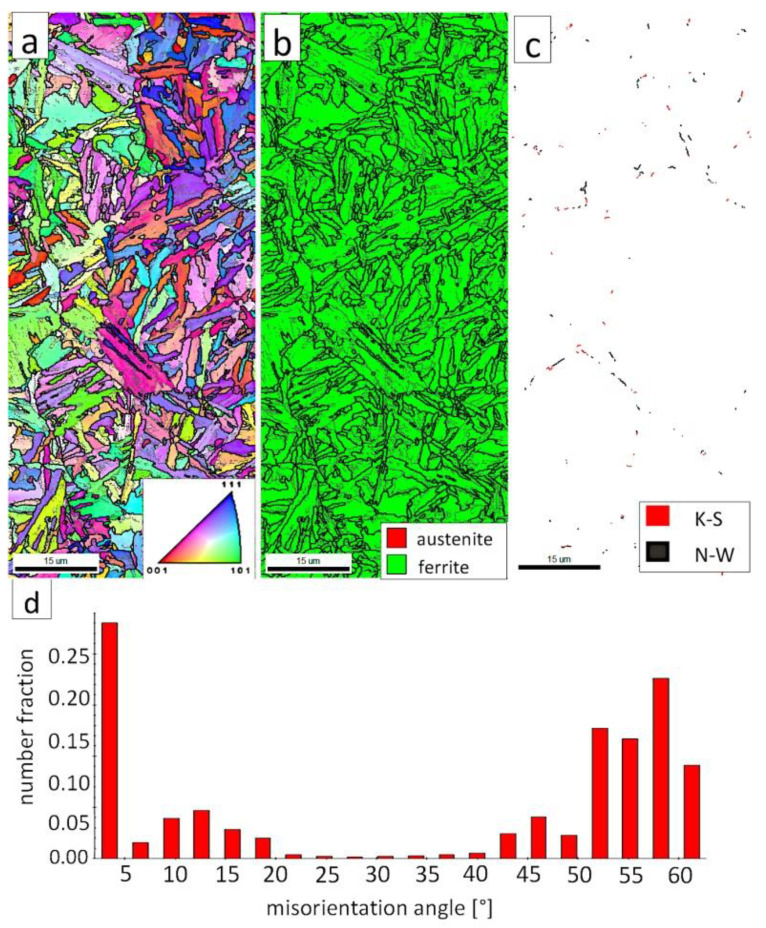
(**a**) IPF Map of the LTHAZ. (**b**) Phase Distribution Map of the LTHAZ. (**c**) Phase distribution map with indicated boundaries of N-W (marked in black) and K-S (marked in red) orientations. Ferrite-white, austenite-gray. (**d**) Misorientation angles distribution of the LTHAZ.

**Figure 15 materials-13-04841-f015:**
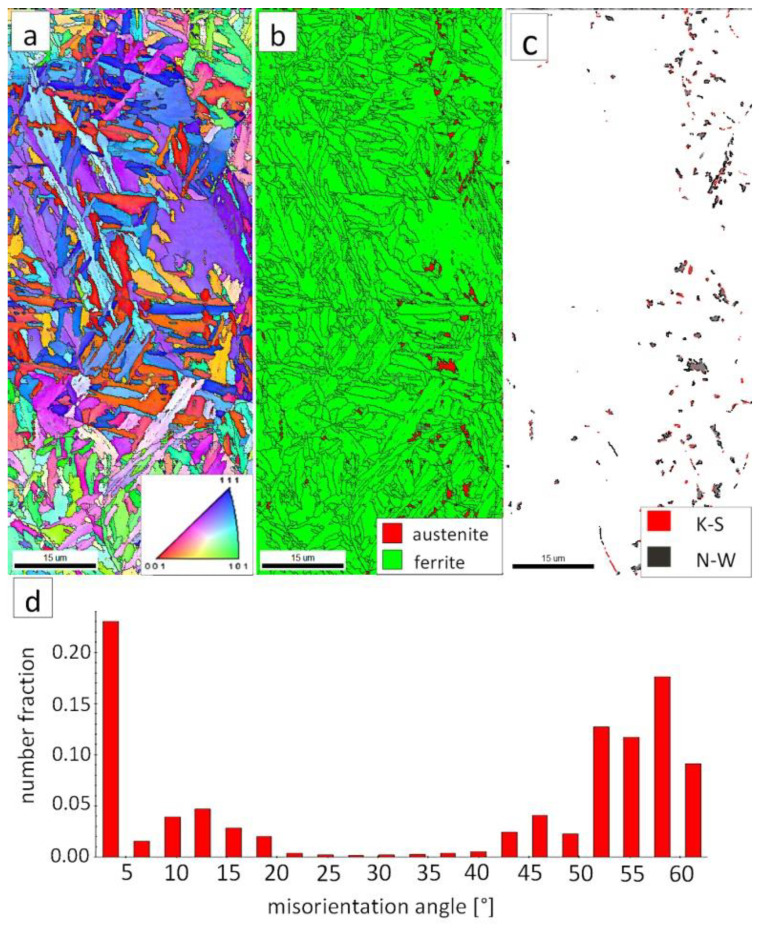
(**a**) IPF map of the BM. (**b**) Phase distribution map of the BM. (**c**) Phase distribution map with indicated boundaries of N-W (marked in black) and K-S (marked in black) orientations. Ferrite-white, austenite-gray. (**d**) Misorientation angles distribution of the BM.

**Figure 16 materials-13-04841-f016:**
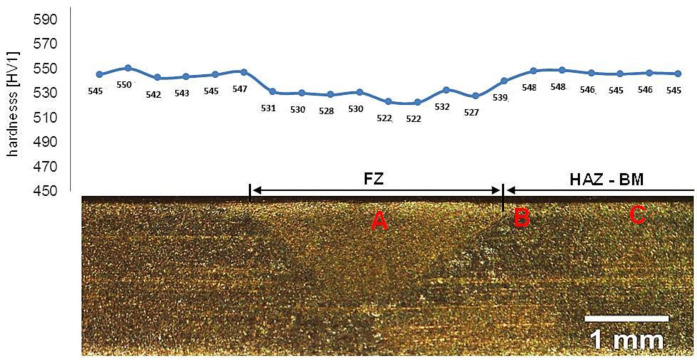
Macroscopic view of welded joint of 55Si7 steel after post welding heat treatment along with the hardness distribution and areas of structural investigations. Etched by saturated picric acid solution and nital, LM. Due to the character of the hardness distribution and the preliminary structure assessment, three areas were selected for further investigations: (**A**) corresponding to FZ, (**B**) corresponding to HAZ, and (**C**) corresponding to BM.

**Figure 17 materials-13-04841-f017:**
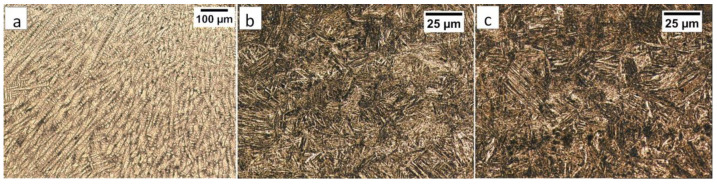
(**a**) The columnar, dendritic solidify structure in the area “A” (marked in Figure 16). Etched by saturated picric acid solution, LM. (**b**) The structure of the Fusion Zone in the area “A” (marked in Figure 16). Lath morphology of bainite. Etched by nital, LM. (**c**) The structure of the HAZ in the area “B” (marked in Figure 16). Lath morphology of bainite. Etched by nital, LM.

**Figure 18 materials-13-04841-f018:**
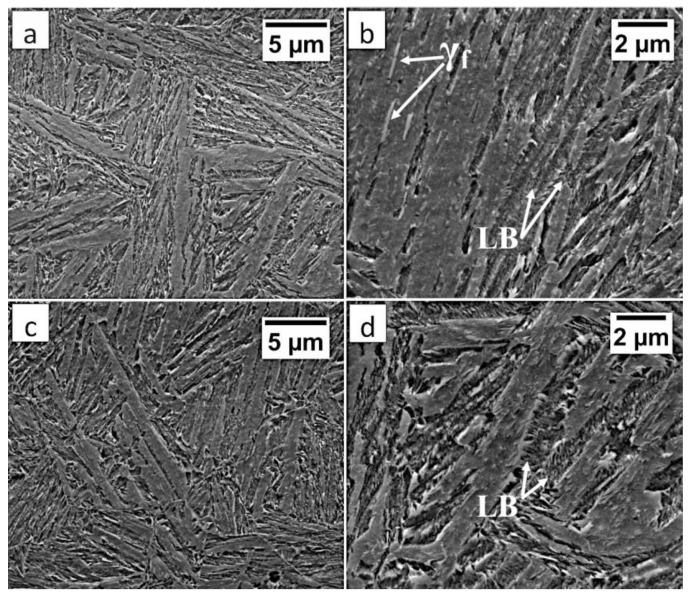
(**a**) The structure of the FZ in the area “A” (marked in Figure 15). Lath morphology of bainite. (**b**) The structure of the FZ in the area “A” (marked in Figure 16). Bainitic ferrite laths and filmy austenite (γ_f_), and lower bainite (LB). (**c**) The structure of the HAZ in the area “B” (marked in Figure 16). Lath morphology of bainite. (**d**) The structure of the HAZ in the area “B” (marked in Figure 16). Degenerated upper bainite and lower bainite (LB). Etched by nital, SEM.

**Figure 19 materials-13-04841-f019:**
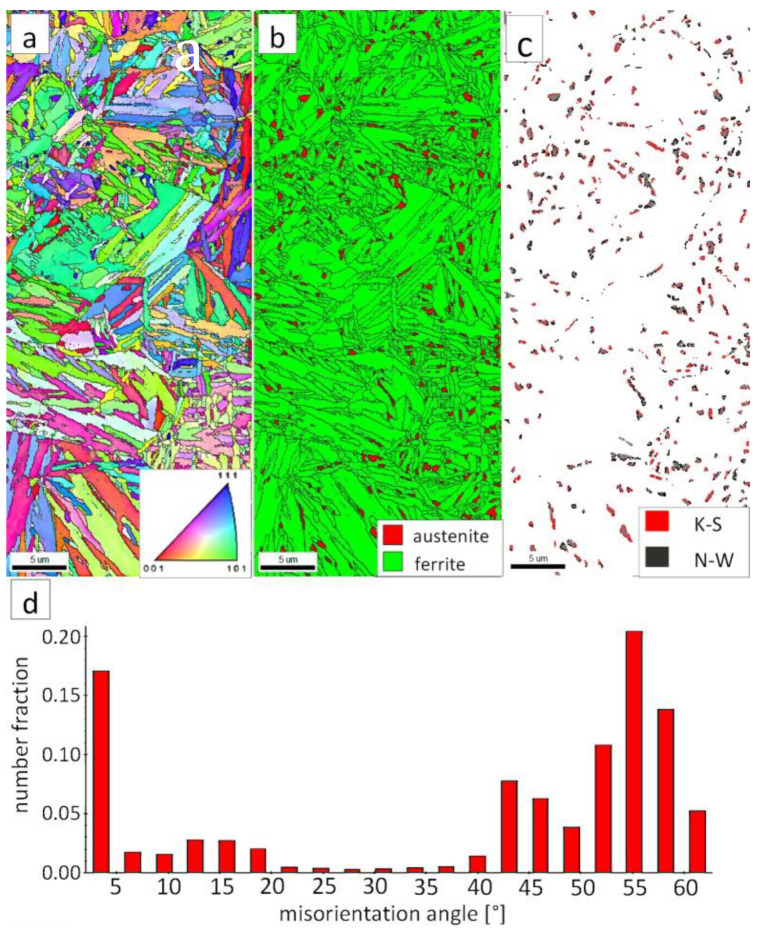
(**a**) IPF Map of the FZ. (**b**) Phase distribution map of the FZ. (**c**) Phase distribution map with indicated boundaries of N-W (marked in black) and K-S (marked in red) orientations. Ferrite-white, austenite-gray. (**d**) Misorientation angles distribution of the FZ.

**Figure 20 materials-13-04841-f020:**
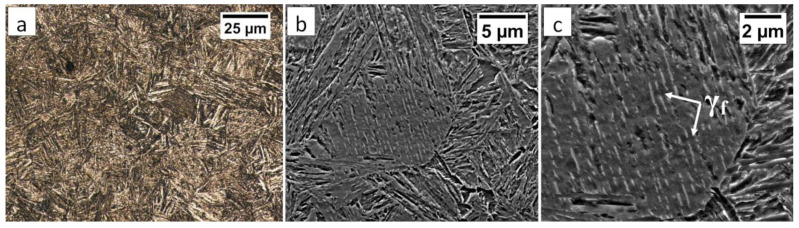
(**a**) The structure of the BM in the area “C” (marked in Figure 16). Lath morphology of bainite. Etched by nital, LM. (**b**) The structure of the BM in the area “C” (marked in Figure 16). Lath morphology of bainite. Etched by nital, SEM. (**c**) The structure of the BM in the area “C” (marked in Figure 16). Bainitic ferrite and filmy austenite (γ_f_). Etched by nital, SEM.

**Figure 21 materials-13-04841-f021:**
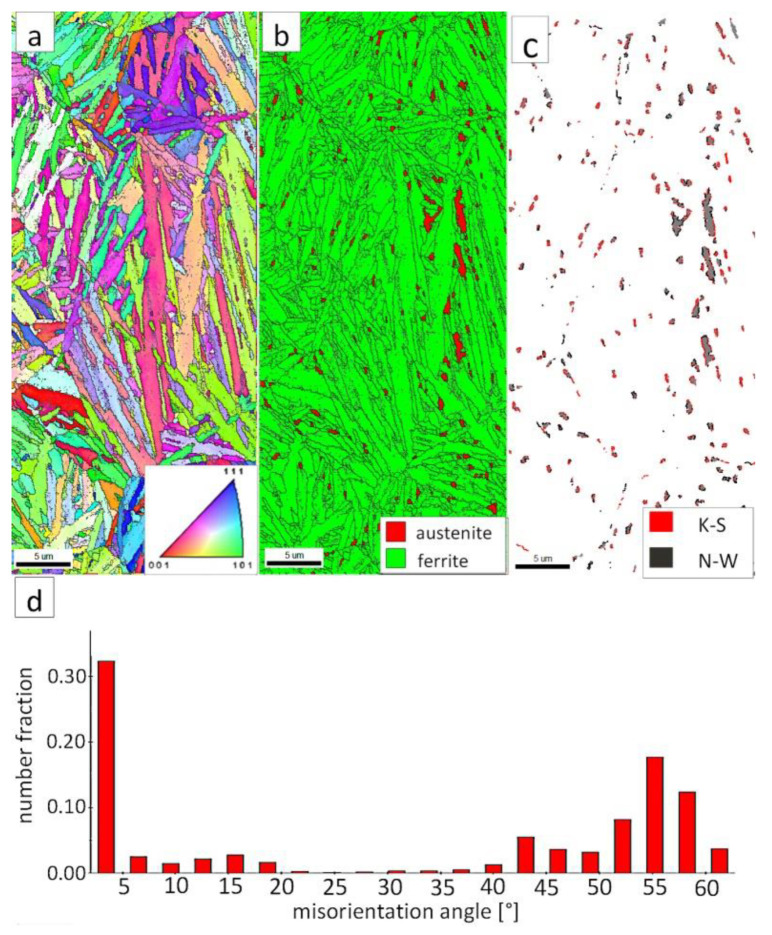
(**a**) IPF map of the BM. (**b**) Phase distribution map of the BM. (**c**) Phase distribution map with indicated boundaries of N-W (marked in red) and K-S (marked in blue) orientations. Ferrite-white, austenite-gray. (**d**) Misorientation angles distribution of the BM.

**Figure 22 materials-13-04841-f022:**
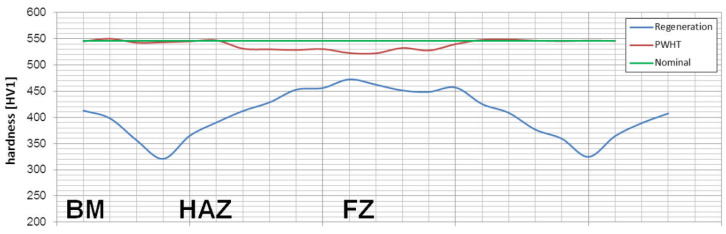
The hardness distributions for performed welding processes of 55Si7 steel. The nominal value corresponds to the base material after isothermal heat treatment before the welding process.

**Table 1 materials-13-04841-t001:** Chemical composition of investigated 55Si7 steel-glow discharge spectrometer GDS500A (Leco Corporation, Saint Joseph, MO, USA).

Chemical Composition (wt.%)						
C	Mn	Si	Cr	Ni	Mo	Fe
0.57	0.69	1.89	0.15	0.19	0.02	balance

**Table 2 materials-13-04841-t002:** Heat treatment parameters of 55Si7 before the welding process.

State	Heat Treatment	Austenitization	Isothermal Annealing	Structure	Hardness (HV1)
as-delivered	softening annealing	-	-	spheroidite in the ferrite matrix, partially lamellar cementite	252.3 ± 2.1
heat treated	isothermal quenching	950 °C/30 min	300 °C/24 h	nanostructured bainite	546.5 ± 2.5

**Table 3 materials-13-04841-t003:** Parameters of the welding process, regeneration technique, and post welding heat treatment.

**Welding Process by TIG Method**									
**State**	**Pre-** **Heating**		**Voltage (V)**	**Current (A)**		**Speed (mm/min)**	**Gas Flow (l/min)**	**Heat Input (kJ/mm)**	**Cooling/Regeneration**
as-delivered	300 °C		18.5	85		400	15	0.142	300 °C/5 min/air
heat treated	300 °C		18.5	85		400	15	0.142	300 °C/10 h
**Post Welding Heat Treatment**									
**State**		**Heating**			**Austenitization**			**Isothermal Annealing**	
as-delivered		stop: 650 °C/5 min			950 °C/30 min			300 °C/24 h	

**Table 4 materials-13-04841-t004:** Orientation relationships between ferrite and austenite [24,25] along with the range of misorientation angles used in the EBSD investigations.

Orientation	Parallelism	Misorientation Angle	EBSD Analysis	Designation
Kurdijumov-Sachs	{111}_γ_║{110}_α_ <110>_γ_║<111>_α_	42.85°	42–44°	red
Nishiyama-Wasserman	{111}_γ_║{110}_α_ <112>_γ_║<011>_α_	45.99°	45–47°	black

**Table 5 materials-13-04841-t005:** Comparison of the phase fraction and the fraction of N-W and K-S orientations obtained by the EBSD analysis.

Zone	Designation (Figure 16)	Austenite Fraction (%)	Ferrite Fraction (%)	N-W Fraction * (%)	K-S Fraction * (%)	Reference
Fusion Zone	A	9.3	90.7	9.6	13.9	Figure 7
HTHAZ	B	19.4	80.6	12.6	18.9	Figure 10
HTHAZ	C	26.0	74.0	12.8	21.3	Figure 11
LTHAZ	D	0.0	100	0.5	0.3	Figure 14
base material	E	1.4	98.6	2.7	1.2	Figure 15

* analysis of N-W and K-S ORs misorientation angles with an accuracy of ± 1°-described in Table 4.

**Table 6 materials-13-04841-t006:** Comparison of the phase fraction and the fraction of N-W and K-S orientations obtained by the EBSD analysis.

Zone	Designation (Figure 16)	Austenite Fraction (%)	Ferrite Fraction (%)	N-W Fraction * (%)	K-S Fraction * (%)	Reference
Fusion Zone	A	3.8	96.2	4.0	4.9	Figure 19
Base Material	C	3.1	96.9	2.3	3.6	Figure 21

* analysis of N-W and K-S ORs misorientation angles with an accuracy of ± 1°-described in Table 4.

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
