# Peer review of "Metallurgical Characterization of Welded Joint of Nanostructured Bainite: Regeneration Technique versus Post Welding Heat Treatment"

_materials, 2020, doi:10.3390/ma13214841_

Round 1

Reviewer 1 Report

This research presents a structure characterization of welded joints of medium carbon 55Si7 grade steel after the welding process with regeneration technique as well as post welding heat treatment (PWHT). The conclusion is the steel with relatively low hardenability (55Si7) is impossible to perform welding with the regeneration technique, and such steels should have been welded supported by PWHT. This paper could be accepted for publication after minor revised. The comments are as follows:

  1. In the abstract, the authors say "it is impossible to perform welding with the regeneration technique, and such steels should have been welded supported by PWHT". The authors should provide more evidence to prove this point.
  2. What is the highlight in this study? The authors should describe this in discussion section.
  3. In discussion section, the authors say "The usage of EBSD methods to identify bainitic morphology in welded joints is reasonable. However, the SEM observations and EBSD investigations should be used complementarily for correct interpretation. Also, suspicions of highly refined phases (e.g. film-like austenite) should be aided by the TEM method". This conclusion is not clear and not sound. The authors should revise it. 

Author Response

Dear Reviewer,

Thank you for your review of our manuscript and a few important comments. We hope that with their help we were able to improve the quality of the article. We have compiled your comments and our responses below:

Point 1-  In the abstract, the authors say "it is impossible to perform welding with the regeneration technique, and such steels should have been welded supported by PWHT". The authors should provide more evidence to prove this point.

Response 1: Thank you for your valuable comment. We agree that this sentence is incorrectly formulated. Therefore, we changed the term "impossible" that raises a lot of controversies. However, we believe that the evidence presented by us in this manuscript (analysis of research results) is sufficient to recommend a more favorable welding method. The conducted extensive material investigations allowed for a clear and unambiguous assessment and comparison of both welding methods used in this research.

Revised to:

Line 27-28. “Based on the structure analysis it was found that steel with relatively low hardenability (55Si7) should be welded using PWHT rather than a regeneration technique.”

Point. 2.  What is the highlight in this study? The authors should describe this in discussion section.

Response 2: Thank you for your comment. The last paragraph of the discussion (Line 590-607) indicated the significance of these investigations in the context of the possibility of welding second-generation NANOBAIN steels (with low manganese and chromium content, and therefore poor weldability). Thus, we believe that we enough highlighted the most important goals of this research in this paragraph.

Point. 3.  In discussion section, the authors say "The usage of EBSD methods to identify bainitic morphology in welded joints is reasonable. However, the SEM observations and EBSD investigations should be used complementarily for correct interpretation. Also, suspicions of highly refined phases (e.g. film-like austenite) should be aided by the TEM method". This conclusion is not clear and not sound. The authors should revise it.

Response 3: Thank you for your suggestion. We revised the conclusion to ensure better readability and understanding. We believe that this conclusion is significant in the context of this research and the characterization of the bainitic microstructure. The correct interpretation of the microstructure investigations results of complex and multiphase bainitic steels is crucial. We also indicated some limitations of the applied research methods (the degree of refinement of the microstructure).

Revised conclusion:

Line 637-642: “The usage of EBSD methods and the SEM observations allows for the identification of the bainite morphology in the welded joints. It needs to be highlighted that the SEM and EBSD methods should be used complementarily to ensure correct interpretation. On the other hand, identification of highly refined phases (e.g. film-like austenite) should be supported by the TEM method.

If the answers are not exhaustive, we will gladly answer the next questions.

Thank you for your consideration

Sincerely

Aleksandra Królicka

Reviewer 2 Report

The study seems to be interesting and can be accepted after a major revision. Kindly, consider the following comments

  1. The English needs to be revised throughout the manuscript.
  2. Revise the abstract as per guidelines.
  3. Replace the range sign with “-”.
  4. Lines 40-42, Explain briefly that why the presence of these elements offer poor weldability.
  5. Line 43: write full name of TIG as it is introduced for the first time in the manuscript.
  6. Line 70: explain A1.
  7. Explain more about the requirement of this study.
  8. Discuss more about the numerical simulations used in this study.

Author Response

Dear Reviewer,

Thank you for your review of our manuscript and a few important comments. We hope that with their help we were able to improve the quality of the article. We have compiled your comments and our responses below:

Point 1-  The English needs to be revised throughout the manuscript.

Response 1: Thank you for your suggestion. The manuscript was proofread. We believe the language was improved in the revised manuscript.

Point. 2.  Revise the abstract as per guidelines.

Response 2: Thank you for your comment. The abstract was shortened following the guidelines.

Point. 3.  Replace the range sign with “-”

Response 3: Thank you for your suggestion. Range sign was replaced with “-“.

Point. 4.  Lines 40-42, Explain briefly that why the presence of these elements offer poor weldability.

Response 4: Thank you for your valuable comments. A brief explanation was added in the revised manuscript.

Line 41-44: “Carbon equivalent (CEV or CE) is the most common tool to determine the weldability of steels. CEV is strictly dependent on the chemical composition of the steel, and weldability is significantly reduced by an increased content of chemical elements such as C, Mn, Cr, Mo, V, Ni, and Cu.”

Point. 5.  Line 43: write full name of TIG as it is introduced for the first time in the manuscript.

Response 5: Thank you for your comment. Revised in the submitted manuscript (Line 48).

Point. 6.  Line 70: explain A1

Response 6: The explanation was added in the revised manuscript (Line 75).

Point. 7.  Explain more about the requirement of this study.

Response 7: The requirements of this study were described in the Introduction and Materials and Methods sections. We assumed the following requirements for these investigations:

  • The parameters of the regeneration technique aimed at obtaining a bainitic structure in the entire area of the welded joint (regeneration at 300 °C, the same as for isothermal heat treatment of the tested steel);
  • PWHT was designed in a similar to isothermal heat treatment 55Si7 steel with a nanobainitic structure.
  • The welding process parameters were the same for PWHT and the regeneration technique to direct comparison.
  • The same preheating was used for both methods.
  • The reference material was 55Si7 steel after isothermal heat treatment with a nanobainite structure.
  • The comparison of welded joints to the reference material included both hardness measurements and an evaluation of the microstructure morphology.

Point. 8.  Discuss more about the numerical simulations used in this study

Response 8: Thank you for your suggestion. The result of the numerical simulation of thermal cycles during welding was described to the detailed determination of the welding process. The numerical simulation is not considered as research results because it supports the description of the welding process. However, as suggested, we discussed the simulation conditions and their results.

Revised:

Line 143-146: “Parameters of the welding process of the numerical simulation were similar to the experimental welding process (Table 3). Preheating was also determined. A modeled sheets used in the numerical simulation was similar to the sheets used in experimental welding processes (2x80x80 mm and 2x100x150 mm).”

Line 148-156:Based on the heat cycles simulation it was found that the measuring points Point 1, Point 2, and Point 3 corresponded to the fusion zone where solidification occurred. The High-Temperature Heat-Affected Zone (HTHAZ) corresponded to the areas where the Tmax was higher than the A3 temperature (Point 4, Point 5, and Point 6). Thus, complete recrystallization of the structure occurred in the HTHAZ regions. However, point 7 corresponded to partial recrystallization (Tmax was in the range between temperatures A1 and A3). This zone was located between the HTHAZ and LTHAZ. Point 8 and Point 9 corresponded to LTHAZ where Tmax did not exceed temperature A1. While Point 10 corresponded to the base material due to the lack of significant influence of the heat-affected related to the welding process.”

If the answers are not exhaustive, we will gladly answer the next questions.

Thank you for your consideration

Sincerely

Aleksandra Królicka